# Depressive Symptoms in Fathers during the First Postpartum Year: The Influence of Severity of Preterm Birth, Parenting Stress and Partners’ Depression

**DOI:** 10.3390/ijerph19159478

**Published:** 2022-08-02

**Authors:** Francesca Agostini, Erica Neri, Federica Genova, Elena Trombini, Alessandra Provera, Augusto Biasini, Marcello Stella

**Affiliations:** 1Department of Psychology “Renzo Canestrari”, University of Bologna, 40127 Bologna, Italy; erica.neri4@unibo.it (E.N.); federica.genova@unibo.it (F.G.); elena.trombini@unibo.it (E.T.); alessandra.provera3@unibo.it (A.P.); 2Donor Human Milk Bank Italian Association (AIBLUD), 20126 Milan, Italy; augustoclimb@gmail.com; 3Pediatric and Neonatal Intensive Care Unit, Maurizio Bufalini Hospital, 47521 Cesena, Italy; marcello.stella@auslromagna.it

**Keywords:** perinatal depression, fathers, preterm birth, severity of prematurity, ELBW, VLBW, parenting stress, partner’s influence

## Abstract

Although preterm birth constitutes a risk factor for postpartum depressive symptomatology, perinatal depression (PND) has not been investigated extensively in fathers of very low (VLBW) and extremely low birth weight (ELBW) infants. This study explored paternal depression levels at 3, 9, and 12 months of infant corrected age, investigating also the predictive role played by the severity of prematurity, maternal and paternal PND levels, and parenting stress. We recruited 153 fathers of 33 ELBW, 42 VLBW, and 78 full-term (FT) infants, respectively. Depression was investigated by the Edinburgh Postnatal Depression Scale (EPDS) and distress by the Parenting Stress Index-Short Form-PSI-SF (Total and subscales: Parental Distress, Parent–Child Dysfunctional Interaction, and Difficult Child). ELBW fathers showed a significant decrease (improvement) in EPDS, total PSI-SF, and Parental Distress mean scores after 3 months. Paternal EPDS scores at 12 months were significantly predicted by VLBW and FT infants’ birth weight categories, fathers’ EPDS scores at 3 and 9 months, Parent–Child Dysfunctional Interaction subscale at 3 months, and Difficult Child subscale at 9 months. This study strengthens the relevance of including early routine screening and parenting support for fathers in perinatal health services, with particular attention to fathers who might be more vulnerable to mental health difficulties due to severely preterm birth.

## 1. Introduction

Preterm birth occurs when infants are born before the 37th gestational week [1] and constitutes an important risk factor for both the survival, health, and development of the newborn. The risk of negative sequelae especially emerges when preterm delivery occurs in “extreme” conditions, such as when gestational age is lower than 32 weeks (very preterm, VPT), or birth weight is less than 1500 g (very low birth weight, VLBW) or even less than 1000 g (extremely low birth weight, ELBW) [2,3].

Prematurity represents a stressful and potentially traumatic event for parents, who might experience feelings of grief, guilt, anxiety, hopelessness, and persistent concerns about their infant’s condition [4]. For this reason, the transition to parenthood after a preterm birth can be complex for both mothers and fathers, and challenges (e.g., parental post-traumatic stress reactions, difficulties in parent–infant bonding, concerns related to preterm infants’ atypical appearance) might also continue after infant discharge [5].

Many studies have focused on the description of main psychological reactions in terms of distress, adjustment difficulties, depression, anxiety, and trauma-related symptoms [6,7,8]. Among these, depressive symptomatology represents one of the most frequent expressions of emotional difficulties after preterm birth.

Depressive symptoms may interfere with the ability of the parent to be emotionally available and sensitive to the infant’s needs, therefore there is an increased risk of negative consequences on the parent–infant relationship and preterm infant’s development [9,10].

Several studies have investigated the presence of depressive symptoms in parents after preterm birth, mostly in mothers [11,12]. The systematic review and meta-analysis by de Paula Eduardo et al. [13] highlighted a higher risk for postnatal depression (PND) among mothers of preterm infants in assessments up to 24 weeks after birth and especially in the early postpartum period, even if some methodological discrepancies were recognized among the studies, (e.g., poor control of confounding variables, lack of control group). However, no similar reviews exist for postnatal depression in preterm infants’ fathers. Particularly, perinatal depressive symptoms in fathers may increase anxiety and feelings of inadequacy related to the assumption of the parental role, hostility, outbursts of rage and aggressive behaviors, social isolation, compromising their ability to support their partner, and the infant [14]. Therefore, it is relevant to deepen the knowledge of paternal affective states after preterm birth. To our knowledge, only some studies have assessed the levels and rates of depressive symptoms in fathers, other than related risk factors, considering different time periods of assessment and heterogeneous samples of preterm infants [15,16].

Regarding the first postpartum weeks, which often correspond to the period of infant hospitalization, Petersen and Quinlivan [12] found no differences in depression scores between 928 full-term (FT) and 72 preterm (PT) fathers at 6 weeks postpartum, even if their PT sample did include different degrees of prematurity (both lower/upper than 2500 g). Cajiao-Nieto et al. [15] assessed at 3 and 20 days after birth depressive symptoms in 51 PT fathers and 33 FT fathers, finding that PT fathers had higher depression scores only at 3 days and that infant’s appearance and behavior and parental role alteration (the perception of having a fragile and less responsive infant, as well as having more barriers to participate to care activities and to assuming paternity, respectively) were the most critical aspects contributing to a higher risk of depression. Nevertheless, the preterm sample included together moderate and late preterm babies (ranging from 32 to 36 gestational weeks). These emerging findings seem to suggest that the first days after a preterm birth represent the most critical period for fathers’ psychological adaptation. Within this research line, Candelori et al. [16] investigated the rates of depression in 32 couples of PT infants’ parents, between 10–20 days after birth; the preterm sample included all the infants under 37 gestational weeks (of which 15.6% ELBW and 25% VLBW). Results showed that the prevalence of parents above the risk threshold for depression was high both for mothers (68.5%) and fathers (37.5%). Helle et al. [17] analyzed one month postpartum the prevalence, risk, and predictors for postnatal depression (PND) in a sample of 230 families, of which 119 FT babies’ parents and 111 VLBW preterm infants’ parents. Results showed a significantly higher risk for PND in VLBW mothers compared to FT ones; VLBW fathers showed a similar trend, even if scores were overall lower compared to VLBW mothers. In sum, the birth of a VLBW infant resulted to be the most relevant risk factor for PND. Finally, Winter et al. [18] assessed 237 VPT babies’ fathers at 38 days after birth for depression, while their babies were still hospitalized and found a depression rate of 16.9%, with no control group included. These studies seem to confirm an increased risk for the occurrence of PND in fathers in the first weeks after preterm birth.

Globally, these findings, given the differences among the studies in terms of research design and methodology (such as preterm samples including different grades of prematurity), as well as the paucity of research on fathers, call for further confirmation.

Very few studies have further explored the depressive symptomatology in preterm infants’ fathers across the first postpartum year. Specifically, two studies have considered the first 6 months after birth. Pace et al. [19] longitudinally investigated depression trajectories in VPT infants’ parents (113 mothers, 101 fathers) compared to FT parents (117 mothers, 110 fathers), with assessments every 2 weeks for the first 12 weeks after birth and at 6 months postpartum. Both VPT mothers and fathers showed a reduction in mean scores and rates of depression symptoms across the first 12 weeks; however, they continued to show higher rates compared to FT parents both shortly after birth (mothers: 40% vs. 6%; fathers: 36% vs. 5%) and at 6 months postpartum (mothers: 14% vs. 5%; fathers: 19% vs. 6%). Ouwendijk et al. [20] investigated mental health in parents (57 mothers, 51 fathers) of VPT infants, at 3 and 6 months of corrected age. VPT infants’ mothers were more likely to experience symptoms and be at risk for a clinical depression disorder than mothers of a reference group from a Dutch population, while preterm infants’ fathers did not manifest an increased risk for symptoms of depression.

Three other studies longitudinally assessed paternal depressive symptomatology up to 12 months after birth. McMahon et al. [21] explored depression trajectories in 100 fathers of 125 VPT infants shortly after birth, at 3, 6, and 12 months postpartum: while 82% of VPT fathers showed persistently low symptoms across the first year, 18% of them exhibited high depressive symptoms which persisted over time. The authors concluded that being a father of a VPT infant increases the risk for a chronic course of depressive symptoms. Vriend et al. [22] investigated levels of depression at 1 and 12 months in VPT infants’ parents, finding that rates with high symptoms tended to decrease across time in both mothers and fathers; regarding the latter, the rate of depressive symptoms decreased from 9% to 4% from 1 month to 12 months. Additionally, Genova et al. [11] longitudinally explored PND trajectories, specifically at 3, 9, and 12 months postpartum, but they compared different conditions of the severity of prematurity and FT: 38 parental couples of ELBW, 56 for VLBW, and 83 FT ones. At 3 months, ELBW parents showed higher PND levels compared to FT and VLBW parents; at the same time, they showed a greater symptom reduction over the first year. Regarding parental role, ELBW mothers, but not fathers, exhibited higher depressive symptoms at 3 months and a higher reduction in symptomatology compared to VLBW and FT groups.

According to those few studies, it emerges that depressive symptoms in preterm infants’ fathers have been poorly investigated over the first postpartum year. This trend is consistent with the more general literature on perinatal psychopathology on normative samples, that has extensively investigated maternal psychological states, while only recently showing a growing interest in the exploration of fathers’ emotional issues [23]. Given the paucity of the literature, a need comes to light to further investigate fathers’ depressive symptoms in the long-term by also considering specific contexts, such as preterm birth. Moreover, as the few mentioned studies focused only on VPT or VLBW conditions, research should make an effort to better explore the influence of the severity of prematurity and the role of potential influencing variables.

In this sense, one of the psychological dimensions experienced by parents after preterm birth, and that could be related to depressive symptomatology, is a high level of stress (i.e., often referred to as psychological distress, parental stress, parenting stress, etc.). Several studies have indeed found a moderate–high amount of stress experienced by preterm babies’ mothers and fathers during baby hospitalization and also after discharge [24,25,26,27]. A systematic review highlighted that significant sources of high stress in fathers in NICU are represented by alteration of the parental role (e.g., limited or denied access to infant’s care, impaired opportunities to establish emotional bonding), infant appearance (e.g., perception of the infant as fragile, less responsive, more irritable), characteristics of NICU environment (e.g., intensity of sights and sounds of the NICU), staff communication (e.g., unsatisfying access to regular information about infant’s health and care) [28]. Additionally, a higher level of stress was related to the severity of prematurity (such as VPT, ELBW) [25,26,29]. Among all these factors, the meta-analysis by Caporali et al. [30] seems to confirm that, during infant hospitalization, the biggest source of stress for both mothers and fathers is represented by parental role alteration, partially independent of the baby’s characteristics, birth weight, gestational age at birth and newborn comorbidities. That would mean that it is the traumatic experience of having the baby hospitalized in the NICU that contributes the largest to the stress experienced. The meta-analysis also highlights that the stress tends to be higher for mothers compared to fathers and this finding could be explained by the different involvement in the parental role: while mothers tend to be involved as the primary caregiver, spending more time in NICUs, fathers often resume job engagement before the end of hospitalization, perceiving, therefore, less parental stress. In this sense, Schmoker et al. [27] explored the levels of stress in preterm infants’ parents during the first postpartum year and found different patterns of symptoms between mothers and fathers (decrease across time in mothers, increase between 6 and 12 months in fathers), highlighting the need to deepen the distinction according to the parental role.

Another factor potentially contributing to a higher risk of depression in fathers might be the level of mothers’ depressive symptomatology. In general, the literature focusing on perinatal depression in parents has frequently analyzed the relationship between maternal and paternal depressive symptoms, often finding a significant association in terms of correlational and predictive analyses [31,32]. According to Vismara et al. [33], the onset of depressive symptoms in first-time fathers and mothers was influenced by their own levels of anxiety and parenting stress as well as by the presence of depression in their partners. The study by Neri et al. [7] has further explored the characteristic of this mutual relationship in the context of preterm birth, highlighting the relevance of considering severity of prematurity. In fact, results showed that a reciprocal influence between partners was significant for VLBW infants’ parents, specifically maternal depressive symptoms at 3 months contributed to paternal depressive ones at 9 months, but this did not happen for the ELBW group.

Based on all these premises, a longitudinal study was developed with the aim of better understanding the occurrence of depressive symptoms in fathers during the first postpartum year and the relationship between symptomatology, severity of prematurity, parenting stress, and mothers’ depressive symptoms.

The first aim of the study was to investigate whether depressive symptoms in fathers differed according to categories of birth weight during the first postpartum year; based on the evidence of the literature, we expected to find higher levels of depressive symptoms in correspondence to a more severe preterm birth (ELBW).

A second aim was to investigate if the level of parenting stress in fathers was different according to the severity of prematurity during the first postpartum year; according to the previous literature, we expected a higher score in more severe preterm babies’ fathers across time.

A third aim was to identify which variables could better predict fathers’ perinatal depressive symptoms at 12 months after birth. We specifically aimed at exploring the role played by the severity of prematurity, depressive symptoms in fathers and mothers at 3 and 9 months after birth, and levels of paternal parenting stress.

## 2. Materials and Methods

### 2.1. Study Design and Participants

This study was part of wider longitudinal research aimed at assessing the parental affective states and infants’ development from 3 to 12 months postpartum after preterm birth.

Families were recruited at the Neonatal Intensive Care Unit (NICU) of Bufalini Hospital (Cesena, Italy) during the period between April 2013 and December 2015; at the same time, the study was presented at the antenatal classes held at Health Services in the same town, in order to recruit a control group composed by parents of healthy FT infants. Eligible participants received complete information about all the aspects of the research from a member of the research team at the moment of NICU discharge or during antenatal classes. Exclusion criteria for all participants were: absence of fluency in Italian language, presence of previous or present psychiatric illness, presence of infants’ chromosomal abnormalities, cerebral palsy, malformations, fetopathy, severe complications (leukomalacia, hydrocephalus, Intraventricular hemorrhage-grades III–IV). In case of twin birth, only the first-born one was included. Infants were monitored during the first year and in case of delays or severe complications they were excluded from the study.

At the end of the recruitment, the sample included 153 participants: 78 were fathers of FT infants, with a birth weight > 2500 g and gestational age > 36 weeks (FT group) and the remaining 75 were fathers of PT infants. According to infant birth weight, 42 fathers were included in VLBW group (birth weight between 1000 and 1500 g) and 33 in ELBW group (birth weight < 1000 g).

The study was approved by the Ethical Committee of the Department of Psychology (University of Bologna) before its start.

### 2.2. Procedure

The assessments took place at 3 months (T1), 9 months (T2), and 12 months (T3) postpartum (corrected age for PT infants) at Developmental Psychodynamic Laboratory (Department of Psychology, University of Bologna, Cesena). At first assessment, a psychologist, blind to infant birth weight, gave all parents a written informed consent to sign and asked them to complete an ad hoc questionnaire regarding socio-demographic and infant clinical information. In addition, during all steps of assessment (T1, T2, and T3) the same psychologist met parents and administer two self-report questionnaires to evaluate the presence of depressive symptomatology in both fathers and mothers and the levels of paternal parenting stress.

### 2.3. Measures

The Edinburgh Postnatal Depression Scale (EPDS) [34] is the most widely used self-report questionnaire for the screening of perinatal depressive symptomatology in both women and men [35]. It is composed of 10 items that investigate the presence of perinatal depressive symptoms in the previous 7 days. Items are scored from 0 to 3 points, and the total EPDS score ranges from 0 to 30, with higher total scores indicating higher levels of depressive symptomatology. A validated Italian version of EPDS questionnaire is available for the assessment of PND for both mothers [36] and fathers [37]. The internal consistency has been demonstrated to be good in both the maternal (Cronbach’s alpha 0.78) and paternal (Cronbach’s alpha 0.83) versions. A cut-off score of ≥10 for women and ≥13 for men has been suggested for the identification of clinically relevant symptoms of postpartum depression [36,37].

The Parenting Stress Index-Short Form (PSI-SF) [38] is a 36-item self-report questionnaire investigating stress specifically associated with parenting on a 5-point Likert scale. It provides a total score and 3 partial scores according to 3 subscales: parental distress (perception of difficulties in parental role related to feelings of being overwhelmed, trapped, and frustrated by parental responsibilities at the expense of other aspects of life); parent–child dysfunctional interaction (difficulties in the interaction with the child related to feelings of unsatisfaction, sensation of not being appreciated and sought by the infant); difficult child (difficulties tied to specific infant characteristics, perceived as irritable, moody, agitated, hyperactive, etc.). Scores of the 3 subscales range from 12 to 60 and the total score (the sum of 3 subscales) from 36 to 180, with higher scores associated with more severe stress symptoms. A Total PSI-SF score ≥ 90 (or above the 90th percentile) is considered to detect individuals with significant level of distress, as indicated by the Italian version validated by Guarino et al. [39]. The questionnaire has good overall psychometric proprieties, with an internal reliability coefficient (Cronbach’s alpha) of 0.91 for the total score, and >0.80 for the 3 subscales.

### 2.4. Statistical Analysis

All statistical analyses were carried out using the IBM SPSS statistical package version 25.0 (IBM, Armonk, NY, USA).

Pearson’s chi-square test and univariate ANOVA were run to verify the homogeneity regarding sociodemographic and clinical variables among ELBW, VLBW, and FT groups. In case of non-homogeneity, we considered the possibility to include those variables in subsequent analyses.

In line with our first aim, we ran repeated measures univariate analysis of variance to compare fathers’ depressive symptoms as a function of birth weight (ELBW, VLBW, FT) and time of assessment (3, 9, and 12 months). Bonferroni’s post hoc analyses were used for comparison within and between groups. Similarly, in line with our second aim, repeated measures multivariate analysis of variance was performed for exploring the impact of birth weight (ELBW, VLBW, FT) and time of assessment (3, 9, and 12 months) on total and 3 subscale scores of PSI-SF (parental distress, parent–child dysfunctional interaction and difficult child).

According to the third aim, a linear regression model was performed to identify possible predictors for fathers’ EPDS scores at 12 months (dependent variable). Regression models were tested by backward method in order to reduce the risk of Type II error [40]. Selected predictors were: birth weight, fathers’ and mothers’ depressive symptoms (at 3 and 9 months), fathers’ sources of parenting stress (at 3 and 9 months). Given that birth weight variable included more than two conditions, we split “birth weight” into 2 different categorical variables: birth weight 1 (FT vs. VLBW and ELBW fathers), and birth weight 2 (ELBW vs. VLBW and FT fathers).

## 3. Results

### 3.1. Sociodemographic and Clinical Characteristics

Descriptive analyses showed that the three birth weight groups were homogenous in relation to all sociodemographic and clinical variables, except for parity (X^2^_(2)_ = 38.85; *p* < 0.005), and level of education (X^2^_(2)_ = 7.47; *p* = 0.024). Specifically, FT fathers, compared to VLBW and ELBW ones, were nulliparous in a higher percentage, while FT and VLBW fathers had a higher level of education than ELBW ones (Table 1). Given the differences in the distribution of parity and level of education, and that their potential influence on paternal PND was recognized in previous studies [31,41,42,43,44,45], these variables were included in further analyses to control their possible influence.

Additionally, as expected, significant differences among the three groups emerged regarding type of delivery (X^2^_(2)_ = 48.41; *p* < 0.005), multiple birth (X^2^_(2)_ = 26.78; *p* < 0.005), and gestational age (F_(2, 150)_ = 937.63; *p* < 0.005). Specifically, in the FT group, cesarean section delivery and multiple births were less frequent compared to preterm groups, while a lower gestational age was found in preterm groups. Since these variables, along with the differences that emerged, were strictly linked to preterm status and are coherent with group belonging based on birth weight, they were not included in subsequent analyses.

### 3.2. Fathers’ Depressive Symptoms

In line with our first aim, we compared fathers’ EPDS scores among the three birth weight groups, controlling for confounding variables (parity and level of education).

Results showed no significant effect of birth weight on fathers’ EPDS scores (F_(2, 143)_ =0.20, *p* = 0.817), meaning that fathers, independently from the birth weight group, showed similar EPDS mean scores (Table 2).

When the interaction between birth weight and time of assessment was considered, a significant within effect emerged (F_(2, 143)_ = 4.40, *p* = 0.014) (Table 2): ELBW fathers’ scores significantly decreased from 3 months to 9 and 12 months (Bonferroni post hoc *p* = 0.037; *p* < 0.005, respectively) (Figure 1); conversely, no significant differences emerged among EPDS scores of VLBW and FT fathers in the three times of assessment (Figure 1). Moreover, no between-group significant differences emerged at any time at Bonferroni post hoc.

### 3.3. Fathers’ Parenting Stress

No significant differences according to birth weight emerged among the three groups in PSI total score [F_(2, 143)_ = 0.33; *p* = 0.718] nor in any of the PSI-SF subscales: Parental Distress [F_(2, 143)_ = 0.81, *p* = 0.446]; Parent–Child Dysfunctional Interaction [F_(2, 143)_ = 0.12, *p* = 0.950]; Difficult Child [F_(2, 143)_ = 0.051, *p* = 0.950] (Table 2).

Regarding the interaction between birth weight and time of assessment (Table 2), a significant within effect emerged for PSI total score [F_(2, 143)_ = 4.56; *p* = 0.012]: the level of distress in ELBW fathers significantly decreased from 3 months to 9 and 12 months (Bonferroni post hoc: *p* < 0.005; *p* = 0.001, respectively) (Figure 2), while no significant effects were found in case of VLBW and FT fathers (Figure 2). No differences among birth weight groups emerged at any time of assessment.

In the case of PSI subscales, the interaction between birth weight and time of assessment significantly influenced scores for the PD subscale [F_(2, 143)_ = 3.29; *p* = 0.040] (Table 2). According to Bonferroni post hoc analyses, at 3 months ELBW showed significantly higher scores than VLBW group (*p* = 0.044); moreover, ELBW fathers’ scores significantly decreased from 3 to 9 and 12 months (*p* = 0.001; *p* = 0.006, respectively) (Figure 3).

No significant differences emerged when PCDI and DC subscales were considered (Figure 2).

### 3.4. Predictors of Fathers’ Depressive Symptoms at 12 Months

At a preliminary level, to explore possible multicollinearity among variables, we ran correlation analyses considering if EPDS paternal scores at 12 months were associated with mothers’ and fathers’ EPDS and PSI at 3 and 9 months. Results showed that these variables were significantly but moderately correlated (all Pearson’s *r* ≤ 0.70).

Regression analysis showed a significant model [F_(5, 44)_ = 45.523, *p* < 0.005], with an R^2^ Adjusted = 0.648. According to the model, fathers’ EPDS scores at 12 months were significantly predicted by the following variables: birth weight condition (to be a father of a VLBW or FT infant); high fathers’ EPDS scores at 3 and 9 months; high level of distress related to PCDI at 3 months and to DC at 9 months (Table 3). Among these, the predictors with higher β scores were paternal EPDS scores at 3 months, followed by fathers’ EPDS and DC subscale scores, both related to 9 months.

Considering all the predictors included in the model, maternal EPDS scores and fathers’ PD at any time of assessment did not significantly contribute to fathers’ EPDS scores at 12 months, nor did parity or education.

## 4. Discussion

This study intended to explore the occurrence and characteristics of fathers’ PND symptomatology across the first postpartum year, according to the severity of prematurity (ELBW, VLBW, and FT conditions). Additionally, it aimed to shed light on the predictive role of specific variables, at infant (birth weight), paternal (PND and parental distress), and maternal (PND symptoms) levels, on fathers’ long-term depressive risk after preterm birth. This is one of the first research studies focused on the long-term emotional and psychological adaptation of fathers considering the severity of prematurity. Therefore, it will contribute to deepening the knowledge of fathers’ postpartum well-being, given it is recognized as a potential risk or protective factor for child development, family, and parental functioning [46].

The first aim of this study was to investigate the impact of the severity of preterm birth on paternal PND symptomatology during the first postpartum year. A first result seemed to suggest that, as a function of birth weight, significant differences among ELBW, VLBW, and FT groups did not emerge. This result is in line with a recent study [20] reporting comparable levels of PND between VPT and FT fathers, but it is in contrast with our expected results and with a previous investigation [19], showing higher levels of depression in VPT fathers with respect to FT ones. The inconsistency of these findings might be explained by the different timeframe within whom the fathers were compared, and by the characteristics of the included samples. Differently from our study, in fact, the reported previous investigations were conducted within the first six months postpartum, and they included only one specific category of preterm population, without comparing the risk for PND connected to different degrees of prematurity.

Comparable levels of PND symptoms in ELBW, VLBW, and FT fathers at 3, 9, and 12 months postpartum emerged also when we considered the interaction between birth weight and time of assessment. Our results whereby at 3 months postpartum VLBW and FT fathers showed comparable depression levels are in line with previous investigations on PT and FT samples [12,15], but are in contrast with a study on VLBW fathers [17], indicating the need for better clarifying the relationship between severity of prematurity and risk for PND. The lack of significant differences could be explained by the influence of several factors: firstly, men tend to display depressive symptomatology more through externalizing behaviors (i.e., anger attacks, acting outs, addictions, etc.), rather than typically depressive-like responses and the use of screening tools originally developed for mothers (such as the EPDS) might not be appropriate [14,47]; secondly, they tend to see mental health difficulties as a sign of weakness, threatening masculinity, and they may feel it is culturally and socially unacceptable to express them [14,48]; thirdly, especially in cases of atypical and vulnerable conditions, they may suppress or minimize their depressive states for not compromising support offered to their partners [49]. All these factors could have made difficult an accurate identification of the depressive risk after preterm birth, especially if severe. For these reasons, and given the paucity of the literature, this issue needs to be further deepened.

However, in line with recent studies [7,11,19], an interesting trend did emerge that sees ELBW fathers showing high EPDS scores at 3 months, followed by a significant decrease at 9 months and by a stable trend till 12 months, while VLBW and FT fathers’ symptoms remained quite constant across all the first postpartum year. These findings seem to suggest that the risk for PND is related to higher severity of prematurity (ELBW) and that the most challenging period for fathers’ emotional and psychological adaptation is represented by the first months postpartum when the demands tied to the preterm birth are more relevant (infant hospitalization, medical complications, support to the partner, etc.). Additionally, the reduction in depressive symptoms observed in the following months seems to be in line with other studies [11,19,22], allowing prudent hope about the fact that the effects of severely preterm birth could be acute but not chronic.

The second aim of this study was to investigate the impact of the severity of prematurity on parental distress during the first year after birth. As a function of birth weight, we did not find differences (nor at global or subscale levels) among ELBW, VLBW, and FT groups. These results are partially in line with a recent study [20], which observed similar levels of perceived stress between VPT fathers and FT fathers, but more everyday problems related to parenting, spouse relationship, and physical and cognitive domains for VPT fathers in the first 6 months after birth. When we looked at the interaction between birth weight and time of assessment, we found that at 3 months postpartum only, ELBW fathers reported significantly higher mean scores on the PD subscale compared to VLBW ones (FT group reported a lower mean score too, but not significantly). On the contrary, the three groups did not differ at a global level or on PCDI and DC subscales. These findings seem to confirm that the consequences of preterm birth are related to its severity, with a greater psychological burden in terms of stress for fathers of severely preterm infants, particularly in the first months following birth, as evidenced by Ionio et al. [26] and Hames et al. [24]. In addition, they strengthen the empirical observation that the biggest source of stress is represented by parental role alteration, confirming previous studies [28,29,30]. Indeed, the traumatic and unexpected interruption of the process of the transition to parenthood catapults these “extremely preterm” fathers in their parental role, calling them to quickly adapt to the demands of the situation, while they must cope with feelings of fear, helplessness, and uncertainty related to the extreme vulnerability of their babies. These results add further support to the possibility that, given the stressful scenario in which the transition to parenthood happens, fathers of severely preterm babies might feel more overwhelmed by the demands of their parental role in the first postpartum months. Again, the results seem to confirm that, in the following months, “severely preterm” fathers gradually feel less stressed and more able to cope with their parental responsibilities, showing resilient behaviors and adaptability to the parental role.

However, it is noteworthy that the comparable levels of stress observed in our VLBW and FT fathers are not in line with previous longitudinal investigations where: for VPT infants’ fathers an increase in stress levels was observed from 6 to 12 months postpartum [27]; in case of less severe preterm birth (moderately and late preterm infants), higher paternal distress levels were observed particularly at 6 and 12 months [50]. So far, as no previous studies on long-term adaptation to stress in fathers of severely preterm infants have been conducted to be compared to our findings, more research is needed for a better understanding of the course of fathers’ parental distress across the first postpartum year.

For an accurate clinical interpretation of these findings, it is important to consider the specific characteristics of NICU care intervention [51]. Our preterm sample was recruited at the NICU of Bufalini hospital (Cesena), where treatment encompasses modern care principles (e.g., encouraging kangaroo care, early breastfeeding, parental participation in baby care, and unrestricted visiting) and care interventions were provided also after discharge, during follow-up meetings for monitoring infant growth, neurodevelopment, and psychological health. Therefore, these NICU peculiarities could partially explain the risk for PND similar between VLBW and FT fathers and the resilience shown by ELBW fathers (suggested by the reduction in depression and parental distress levels after the first trimester). The fact remains that these findings identify in fathers of severely preterm infants a potential “vulnerable” population, with distinct difficulties, needs, and resources.

The third aim of this study was to explore to what extent fathers’ depressive symptoms at 12 months were predicted by birth weight, depressive symptoms in fathers and mothers, and paternal sources of parenting stress at 3 and at 9 months postpartum. First, birth weight condition significantly predicted higher levels of paternal depressive symptoms at 12 months postpartum in the case of the VLBW or FT condition, but not ELBW. This result is somehow unexpected but is coherent with our ANOVA results whereby the impact of severely preterm birth on fathers’ emotional adaptation seems to be acute rather than chronic, configuring more a reaction of exogenous nature (to the trauma of early birth, to the experience of hospitalization, and to the infant’s vulnerabilities), which tends to gradually go into remission across time, when the risks for infant’s survival and health decrease and fathers are reassured by new developmental skills reached by their infants [52,53]. On the contrary, the EPDS scores of VLBW and FT infants’ fathers seemed to be constant and did not regress as a result of an adaptation to fatherhood during the postpartum year. Given the role that different levels of prematurity might play in paternal depressive symptoms, the monitoring of paternal affective states across time is recommended.

Second, higher levels of paternal depressive symptoms at 3 and 9 months predicted higher EPDS scores at 12 months postpartum. This result is in line with the previous literature on both FT [33] and PT samples [7], highlighting that early depressive risk contributes to higher symptoms in subsequent months and the relevance, therefore, of monitoring the evolution of symptoms [11]. In fact, it is noteworthy that the EPDS score at 3 months resulted from the predictor which most significantly contributed to EPDS scores at 12 months.

Regarding the influence of maternal depressive symptoms at 3 and 9 months on fathers’ depression at 12 months, our results did not show any significant contribution, and this is partially in line with a previous investigation by Neri et al. [7], where a significant role of mothers’ depression was found only for VLBW condition, but not for FT and ELBW groups. Nevertheless, these authors aimed to assess only the reciprocal influence between maternal and paternal depressive symptomatology but did not consider the effects of other predictors such as sources of parenting distress. The inclusion of these variables in the present study led to a more complex model, where the role of each factor is weighted by the presence of the others. This complexity could help to reach a more accurate understanding of paternal experience. However, given the lack of this kind of study, future investigations are recommended.

Specific dimensions of parental distress, that is high PDCI scores at 3 months and high DC scores at 9 months, significantly contributed to fathers’ depressive symptoms at 12 months postpartum. These findings are in line with cross-sectional studies on VLBW and FT fathers [54,55], where high levels of parenting stress tended to be associated with more severe postpartum depressive symptoms; also, they are coherent with the study by Vismara and colleagues [33], suggesting the predictive role of parental distress regarding PND in fathers. It is interesting to note that specific sources of stress predicted depressive symptoms at one year: at 3 months the source was represented by the stress related to the perception of a poor relationship, at 9 months greater stress was related to the infant’s behavior and characteristics. Therefore, these findings open to the possibility that postpartum depressive symptoms might be differentially predicted by specific types of stressful experiences across the first 12 months and this interplay between parenting stress and depression needs to be further explored.

We must acknowledge some limitations of this study. First, the sample should be enlarged, and the three birth-weight groups should reach a similar size for further confirmation of the findings. Second, the EPDS, originally developed for screening in mothers, may not be fully appropriate for detecting gender-related differences in the expression of PND symptoms [56]. Additionally, the EPDS is a self-report tool and, given the well-known limitations of this kind of measure, it should be associated with a structured clinical interview. Third, the presence of anxious symptoms has not been assessed, despite them frequently occurring in comorbidity with depressive symptomatology [57]. Finally, we did not investigate the possible influence of both prenatal factors (such as the frequency of prenatal visits, length of hospitalization before the delivery, etc.), and of NICU care intervention on fathers’ emotional postnatal adaptation.

## 5. Conclusions

To sum up, given the paucity of longitudinal studies on fathers’ psychological adjustment after preterm birth, these results shed new light on this field of research, highlighting the relevance of paying particular attention to the situations where fathers may be more vulnerable to perinatal mental health problems, such as a highly severe preterm birth.

The findings of this study strengthen the clinical relevance of including routine screening programs for fathers in perinatal health services, for identifying those cases that, given a complex interplay between exogenous and endogenous risk factors, are at higher risk for a chronic course. This early identification would enable the implementation of targeted specialist interventions for fathers both for reducing the symptomatology and for supporting their parenting role [48,58]. The inclusion of fathers in assessment programs and interventions might also contribute to reducing a mother-centered bias in the practices of perinatal health services, emphasizing a systemic perspective where the whole family plays an essential role in sustaining infant development and health. Therefore, future studies following these points of reference are highly recommended.

## Figures and Tables

**Figure 1 ijerph-19-09478-f001:**
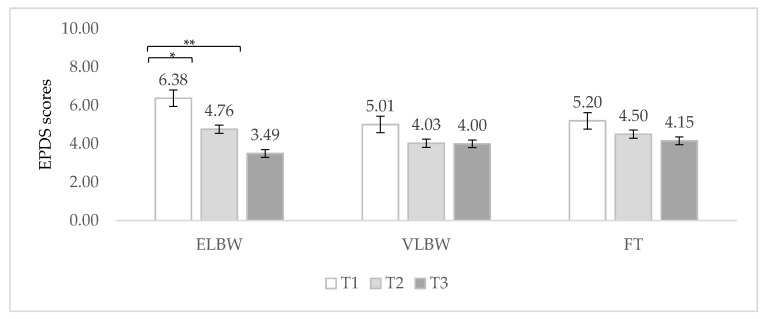
Fathers’ EPDS mean scores related to birth weight and time of assessment. Note. ELBW = extremely low birth weight; VLBW = very low birth weight; FT = full-term; T1 = 3 months; T2 = 9 months; T3 = 12 months. * *p* < 0.05. ** *p* < 0.005.

**Figure 2 ijerph-19-09478-f002:**
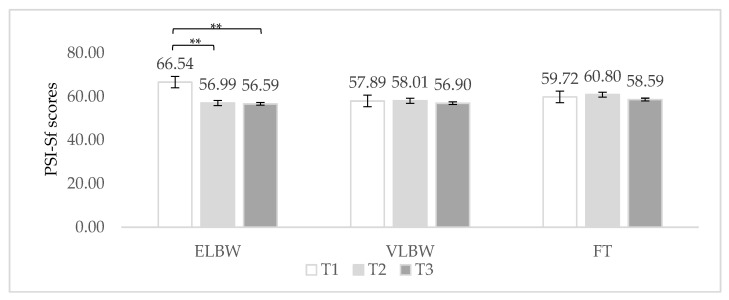
Fathers’ PSI total score related to birth weight and time of assessment. Note. ELBW = extremely low birth weight; VLBW = very low birth weight; FT = full-term; T1 = 3 months; T2 = 9 months; T3 = 12 months. ** *p* < 0.005.

**Figure 3 ijerph-19-09478-f003:**
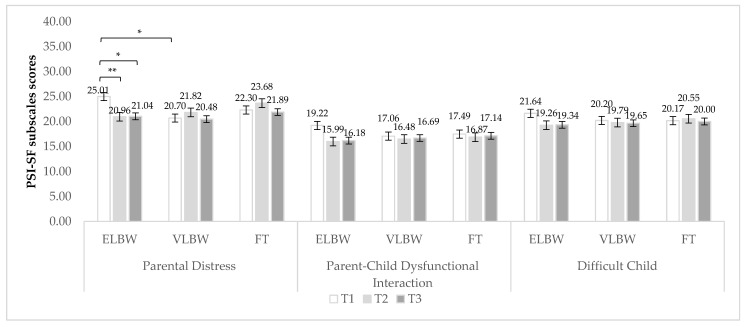
Fathers’ PSI-subscales scores related to birth weight and time of assessment. Note. ELBW = extremely low birth weight; VLBW = very low birth weight; FT = full-term; T1 = 3 months; T2 = 9 months; T3 = 12 months. * *p* < 0.05. ** *p* < 0.005.

**Table 1 ijerph-19-09478-t001:** Sociodemographic and clinical characteristics of father–infant dyads.

	ELBW Group (*n* = 33)	VLBW Group (*n* = 42)	FT Group (*n* = 78)	F/X^2^
Father characteristics				
Paternal age in years ^a^	36.85 (5.2)	37.29 (5.3)	35.63 (5.5)	1.45
Level of education ^b^				7.47 *
Primary/secondary school	15 (45)	9 (21)	17 (22)	
High school/university	18 (55)	33 (79)	61 (78)	
Marital status ^b^				3.66
Married/cohabit	33 (100)	39 (93)	70 (90)	
Other	0 (0)	3 (7)	8 (10)	
Parity ^b^				30.85 **
Nulliparous	21 (70) ^b^	16 (41)	69 (90)	
Multiparous	9 (30)	23 (59)	8 (10)	
Infant characteristics				
Gender ^b^				4.39
Male	16 (49)	28 (67)	37 (47)	
Female	17 (51)	14 (33)	41 (53)	
Gestational age in weeks ^a^	27.55 (2.2)	29.87 (1.5)	40.33 (1.5)	937.63 **
Type of delivery ^b^				48.41 **
Spontaneous	9 (28)	10 (26)	65 (83)	
Cesarean section	23 (72)	29 (74) ^b^	13 (17)	
Multiple birth ^b^				26.78 **
Yes	6 (18)	15 (36)	1 (1)	
Not	27 (82)	27 (64)	77 (99)	

Note. ELBW = extremely low birth weight; VLBW = very low birth weight; FT = full-term. ^a^ Means (and standard deviations in parentheses) for interval data. ^b^ Number (and % in parentheses) for categorical data. * *p* < 0.05, ** *p* < 0.005.

**Table 2 ijerph-19-09478-t002:** Fathers’ mean scores for EPDS and PSI-SF to birth weight and time of assessment.

	Birth Weight			Birth Weight × Time of Assessment									F	
				ELBW			VLBW			FT				
	ELBW (*n* = 33)	VLBW (*n* = 42)	FT (*n* = 78)	T1	T2	T3	T1	T2	T3	T1	T2	T3	Birth Weight	Birth Weight × Time of Assessment
EPDS	4.62 ±0.49	4.88 ±0.61	4.35 ±0.56	6.38 ±0.77	4.76 ±0.69	3.49 ±0.62	5.01 ±0.70	4.03 ±0.63	4.00 ±0.57	5.20 ±0.62	4.50 ±0.56	4.15 ±0.50	0.20	4.40 *
Parental distress (PD)	22.34 ±1.10	21.00 ±0.99	22.62 ±0.88	25.01 ±1.27	20.96 ±1.24	21.04 ±1.33	20.70 ±1.15	21.82 ±1.12	20.48 ±1.20	22.30 ±1.02	23.68 ±1.00	21.89 ±1.06	0.81	3.39 *
Parent–child dysfunctional interaction (PCDI)	17.13 ±0.77	16.74 ±0.70	17.17 ±0.2	19.22 ±0.97	15.99 ±0.88	16.18 ±0.98	17.07 ±0.87	16.48 ±0.80	16.69 ±0.89	17.49 ±0.77	16.89 ±0.71	17.14 ±0.79	0.12	2.65
Difficult child (DC)	20.08 ±0.95	19.88 ±0.86	20.24 ±0.76	21.65 ±1.14	19.26 ±1.09	19.34 ±1.07	20.20 ±1.03	19.80 ±0.99	19.65 ±0.97	20.17 ±0.91	20.55 ±0.87	20.00 ±0.86	0.05	1.54
PSI total score	60.04 ±2.45	57.60 ±2.24	59.70 ±1.98	66.54 ±2.80	56.99 ±2.69	56.59 ±2.95	57.89 ±2.56	58.01 ±2.46	56.91 ±2.70	59.72 ±2.26	60.80 ±2.18	58.59 ±2.39	0.33	4.56 *

Note. ELBW = extremely low birth weight; VLBW = very low birth weight; FT = full-term; EPDS = Edinburgh Postnatal Depression Scale; T1 = 3 months; T2 = 9 months; T3 = 12 months. Values are means ± standard deviations. * *p* < 0.05.

**Table 3 ijerph-19-09478-t003:** Regression model identifying predictors of fathers’ EPDS scores at 12 months.

	T	β	t	*p*
Constant	−3.470		−4.142	<0.001
Birth weight 2	1.314	0.160	3.118	0.002
Fathers’ EPDS scores at 3 months	0.377	0.478	7.459	<0.001
Fathers’ EPDS scores at 9 months	0.264	0.291	4.358	<0.001
Fathers’ PCDI scores at 3 months	0.090	0.139	2.329	0.021
Fathers’ DC scores at 9 months	0.164	0.280	4.398	<0.001

Note. Birth weight 2 = extremely low birth weight fathers vs other fathers (VLBW and FT); PD = Parental Distress subscale; PCDI = Parent–Child Dysfunctional Interaction subscale; DC = Difficult Child subscale.

## Data Availability

Data available on request due to privacy and ethical restrictions.

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
