# Peer review of "Depressive Symptoms in Fathers during the First Postpartum Year: The Influence of Severity of Preterm Birth, Parenting Stress and Partners’ Depression"

_ijerph, 2022, doi:10.3390/ijerph19159478_

Round 1
Reviewer 1 Report
Review Report
Comments for the author
This manuscript conducted a longitudinal follow-up cross-sectional study: Depressive Symptoms in Fathers During the First Postpartum Year: The Influence of Severity of Preterm Birth, Parenting Stress and Partners’ Depression, which is an interesting issue for identifying the relationship between the father’s psychological health and the severity of preterm birth. While there are several unclear descriptions in this manuscript should be concerned and require greater details and re-organized.
1. In abstract, Line 26, the description is unclear: “…predicted by VLBW and FT birth weight categories, …” it should be revised as the VLBW and FT birth weight categories of the infants.
2. In abstract, the results mentioned the variable of “Parent-Child Dysfunctional Interaction subscale and Difficult Child subscale”, however, what are meaning of the subscales? since it was not presented in the study purpose, nor indicated in the method section of the abstract.
3. In the introduction (page 2, line54-55 and line 60-61), please cite references to support your statements.
4. In page 4, the method section: 2.1 participants and procedure, please describe the exclusion criteria.
5. In page 5, line 206-212, please have more detailed descriptions such as: who collected the data and did the data collector be blinded to the group assignment?
6. In page 5, 2.3 statistical analysis section, please indicate the statistical software and the version of the statistical analyses.
7. Simple sentence would be better. in page 5, line 251-252, the sentence is suggested to be” According to the third aim, we used a linear regression model (backward method) to examine the possible predictors of fathers’ EPDS scores at 12 months (dependent variable).”
8. Why did the author used generalized estimation equations for repeat measure since you can examine the changes over time in the fathers’ EPDS under adjustment for the potential sociodemgraphic covariates?
9. The Table 2 is unclear. In the column of the body weight in each group including ELBW, VLBW, and FT, did the values present as the mean of 3 times measures ? if so, what did these values imply?
10. In the Table 4, the constant variable and value can be deleted.
11. The discussion section was very comprehensive, while it was suggested to merge the second and third paragraphs into one paragraph.

Reviewer 3 Report
I thank the editor and authors for the opportunity to review a manuscript. I congratulate authors on an interesting and unique research project. First of all, prematurity belongs to the main medical, psychological and socio-economical problem worldwide. Secondly, parental depression has been studied by many researchers focusing on maternal depression so far. This study explored the occurrence and characteristics of fathers’ postanal depression across the first postpartum year in terms of the severity of prematurity. Authors also evaluated a predictive role of specific variables, such as infant, paternal and maternal outcomes on fathers’ long-term depressive risk after a preterm birth. The current findings add to a growing body of literature on paternal depression levels during the first year postpartum after a preterm birth. The paper has overall a very good technical content and it’s easily readable. I found some of the description of the paper to be less detailed. I offer the following comments
(1) Title: I suggest adding the study design.
(2) Materials and Methods. This section is well written; however, I recommend to start it with the label “Trial design” where you present key elements of study design in addition to the main project. Please add move here information about ethical approval which is placed under the inappropriate header now
(3) Results: start this section with (1) relevant dates, including periods of recruitment, exposure, follow-up, and data collection; (2) numbers of individuals at each stage of study—eg numbers potentially eligible, examined for eligibility, confirmed eligible, included in the study, completing follow-up, and analysed. (3) Consider use of a flow diagram.
Reviewer 4 Report
Thank you for the opportunity to review the manuscript reporting on a study of the influence of preterm birth severity on depressive symptoms in fathers during the first postpartum year.
The research is coherently justified and reported, and appears to have been undertaken rigorously.
I suggest that the layout of Table 2. is refined to assist the reader. The columns are currently close without boarders and as such, it is difficult to differentiate.
A thorough edit of the manuscript by someone who uses English as their first language will enhance reader experience. For example, I note a few minor suggestions:
- Line 71, 'Anyway' - perhaps 'Nevertheless' or ' However' may be a better choice for an academic publication.
- Line 156, delete 'More' and commence the sentence as 'In general'.
- Line 180, reword the first sentence regarding third aim in keeping with the style used for the previous aims. For example, "A third aim was to identify which variables . . . "
Line 198, delete the word 'one' after firstborn.
Line 255, 'we splitted' should be reworded to 'we split'.
Results section, I suggest that all use of the word 'twinning' is replaced with twins, twin, multiple birth or other appropriate term.
Best wishes with your work
